# Positive feedback regulation of *frizzled-7* expression robustly shapes a steep Wnt gradient in *Xenopus* heart development, together with sFRP1 and heparan sulfate

Takayoshi Yamamoto[1]*, Yuta Kambayashi[1], Yuta Otsuka[2], Boni A Afouda[3], Claudiu Giuraniuc[3], Tatsuo Michiue[1,2], Stefan Hoppler[3]

[1]Department of Life Sciences, Graduate School of Arts and Sciences, The University of Tokyo, Tokyo, Japan; [2]Department of Biological Sciences, Graduate School of Science, The University of Tokyo, Tokyo, Japan; [3]Institute of Medical Sciences, The University of Aberdeen, Aberdeen, United Kingdom

**Abstract** Secreted molecules called morphogens govern tissue patterning in a concentration-dependent manner. However, it is still unclear how reproducible patterning can be achieved with diffusing molecules, especially when that patterning concerns differentiation of thin tissues. Wnt is a morphogen that organizes cardiac development. Wnt6 patterns cardiogenic mesoderm to induce differentiation of a thin tissue, the pericardium, in *Xenopus*. In this study, we revealed that a Wnt receptor, *frizzled-7*, is expressed in a Wnt-dependent manner. With a combination of experiments and mathematical modeling, this receptor-feedback appears essential to shape a steep gradient of Wnt signaling. In addition, computer simulation revealed that this feedback imparts robustness against variations of Wnt ligand production and allows the system to reach a steady state quickly. We also found that a Wnt antagonist sFRP1, which is expressed on the opposite side of the Wnt source, accumulates on N-acetyl-rich heparan sulfate (HS). N-acetyl-rich HS concentration is high between the sources of Wnt and sFRP1, achieving local inhibition of Wnt signaling via restriction of sFRP1 spreading. These integrated regulatory systems restrict the Wnt signaling range and ensure reproducible patterning of the thin pericardium.

*For correspondence: tyamamoto@bio.c.u-tokyo.ac.jp

## Editor's evaluation

The study deals with the mechanisms that establish the Wnt gradient combining a mathematical model and experiments considering multiple extracellular components such as receptor and diffusible antagonist. The study revealed that the ligand/receptor feedback enables robust and quick formation of the morphogen gradient and that the diffusible antagonist also plays a role in this process.

## Introduction

Morphogens are secreted molecules that pattern embryonic tissues with concentration gradients that are highest near a localized source and decrease with distance from that source. These molecules are important not only for the embryo, but also for the adult. However, the means by which robustness of morphogen distribution is reliably maintained to ensure reproducible patterning are still debated (*Shilo and Barkai, 2017*).

Distribution of morphogens is influenced by extracellular molecules. Among them, receptors are important since they not only transmit signals to cells, but also trap and locally enrich morphogens on receptor-expressing cells, thereby restricting morphogen distribution to neighboring cells (*Figure 1—figure supplement 1A*). In many cases, receptors also mediate internalization of morphogens into cells and subsequent degradation (*Cadigan et al., 1998*; *Goodrich et al., 1996*). In some cases, receptors regulate morphogen distribution via ligand-dependent receptor expression (*Goodrich et al., 1996*). Mathematical-modeling studies have shown that positive feedback of receptor expression imparts robustness against variations in ligand expression (*Eldar et al., 2003*). In addition to receptors, extracellular molecules, such as heparan sulfate (HS) and secreted antagonists of morphogens, can shape morphogen gradients (*Mii and Taira, 2009*; *Yan and Lin, 2009*).

Wnt morphogen has emerged as a key regulator of vertebrate heart development (*Ruiz-Villalba et al., 2016*). Canonical Wnt/β-catenin signaling restricts differentiation of myocardium tissue (prospective heart muscle) and promotes differentiation of alternative heart tissues, namely pericardium, the prospective lining of the pericardial cavity. Pericardium serves to contain serous fluid in the pericardial cavity, which reduces friction between the beating heart and more tightly fixed surrounding tissues. Pericardium, therefore, needs to be thin, so as to be flexible (*Jaworska-Wilczynska et al., 2016*).

It is still generally unclear which *wnt* genes encode relevant Wnt signals to regulate heart development, with possible differences among vertebrate classes, as well as redundancy in some species (*Mazzotta et al., 2016*; *Ruiz-Villalba et al., 2016*). Therefore, it is generally difficult to study regulation by Wnt signals in vertebrate heart development. However, in *Xenopus* heart development, it has been established that Wnt6 is secreted from ectoderm-derived epidermis to pattern adjacent cardiogenic mesoderm in a concentration-dependent manner (*Lavery et al., 2008a*; *Lavery et al., 2008b*). A relatively thin pericardium differentiates close to the epidermis and a broad myocardium further inside the embryo at a distance from the source of Wnt6 ligand (*Figure 1A*). However, it is still unclear how Wnt6 protein distribution and its signal-active range are regulated to ensure reproducible positioning of pericardium and myocardium in cardiogenic mesoderm.

In early embryos, the range of Wnt8 signaling is precisely regulated by two types of heparan sulfate (HS), N-sulfo-rich HS and N-acetyl-rich HS, and secreted Wnt binding proteins, including Frzb (also known as sFRP3) (*Mii et al., 2017*; *Mii and Taira, 2009*). The distributional range of extracellular Wnt8 protein can be shortened by binding to its receptor and N-sulfo-rich HS. Frzb, which was originally described as a secreted Wnt antagonist, binds to N-acetyl-rich HS. Consequently, Frzb prevents Wnt8 from binding to its receptor and to N-sulfo-rich HS. This extends the Wnt8 protein distributional range. In heart tissue, sFRP1, a secreted antagonist of Wnt6, is expressed in the prospective myocardium region (*Gibb et al., 2013*; *Xu et al., 1998*). We wondered whether mechanisms similar to those that operate in early embryos also regulate distribution of Wnt6 in cardiogenic mesoderm.

The Wnt receptor, Frizzled-7 (Fzd7), is expressed in cardiogenic mesoderm, and is essential for heart development (*Abu-Elmagd et al., 2017*; *Wheeler and Hoppler, 1999*). Expression of *fzd7* is increased by Wnt signaling in *Xenopus* neuroectoderm and human embryonic carcinoma cells (*Willert et al., 2002*; *Young et al., 2014*), but there are no such reports in heart development. In this study, we analyzed regulatory mechanisms of Wnt signaling in differentiation of cardiogenic mesoderm, focusing on extracellular components: Fzd7 receptor, sFRP1, and HS.

## Results

### *fzd7* expression is upregulated by Wnt6 signaling

As cardiogenic mesoderm becomes patterned into peri- and myocardium, initially broad *fzd7* expression becomes restricted within prospective pericardium (*Wheeler and Hoppler, 1999*; *Figure 1—figure supplement 1B*). This pattern of expression reminded us of the expression pattern of the pericardium marker, *gata5*, which is positively regulated by Wnt signaling (*Gibb et al., 2013*). Therefore, we wondered whether expression of *fzd7* was promoted by Wnt signaling in heart development.

To test whether Wnt6 signaling is capable of regulating *fzd7* expression, we injected mRNA encoding an inducible β-catenin (Glucocorticoid Receptor (GR)-fused β-catenin) (*Afouda et al., 2008*) into two-cell stage *Xenopus* embryos (*Figure 1B*). GR-fused β-catenin is translocated into the nucleus with dexamethasone (Dex) addition (*Figure 1C*). We added Dex at the tailbud stage (st. 22–23) and induced Wnt-signal-dependent transcription. *fzd7* expression at stage 32/33 increased throughout the

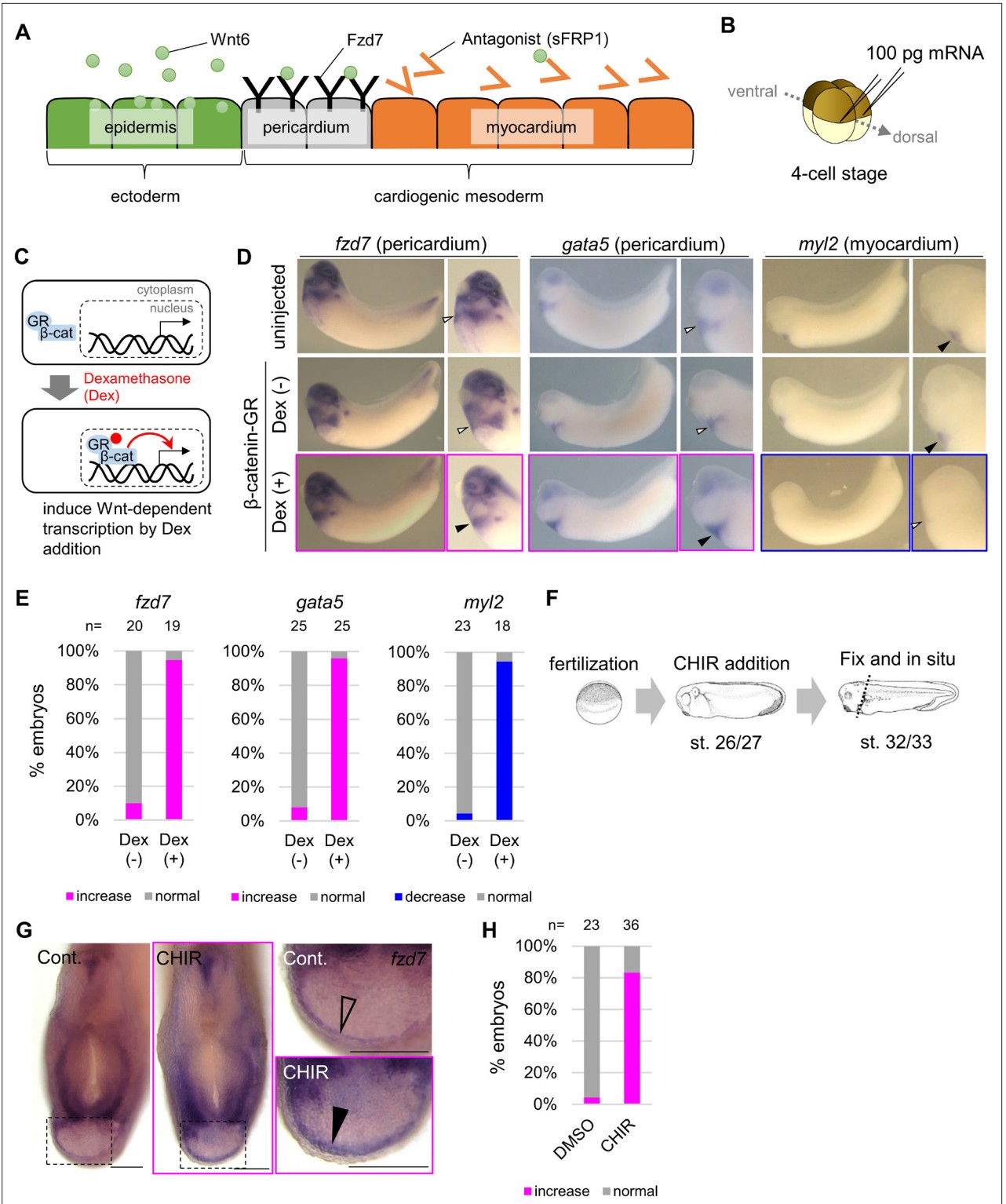

**Figure 1.** Wnt6/β-catenin signaling induces *fzd7* expression during *Xenopus* heart development. (**A**) Schematic figure showing distributions of Wnt6, Fzd7, and sFRP1 in heart development. Wnt6 is secreted from the epidermis (outside the embryo). Its antagonist, sFRP1, is secreted from prospective myocardium (inside the embryo). Expression of the Wnt receptor, Fzd7 becomes localized to the pericardium region as in *Figure 1—figure supplement 1B*. The pericardial cavity will subsequently form between the pericardium and the myocardium. (**B–E**) β-catenin activation increased *fzd7* expression. mRNA encoding an inducible β-catenin protein (β-catenin fused with the hormone-binding domain of the human glucocorticoid receptor (GR)) was injected into two dorsal blastomeres at the four-cell stage (**B**). The GR-fused protein can be translocated into the nucleus with dexamethasone (Dex)

*Figure 1 continued on next page*

*Figure 1 continued*

and induces Wnt/β-catenin signal-dependent transcription (**C**). Dex addition (β-catenin activation) at the tailbud stage (st.22–23) resulted in an increase in *fzd7* and *gata5* expression and a decrease in *myl2* expression (arrowhead, Dex (+)) (**D**) as quantified in E (Fisher's exact test, p=5.8 x $10^{-8}$ (*fzd7*), 1.2 × $10^{-10}$ (*gata5*), 2.1 × $10^{-9}$ (*myl2*)). (**F–H**) Embryos were treated with a Wnt agonist, CHIR99021 (5 µM; control DMSO) from st. 26/27, just before the onset of Wnt6 expression, to st. 32/33 (**F**). The *fzd7* expression area became broader with CHIR treatment (G, arrowhead), but not in DMSO controls (G, open arrowhead), as quantified in H. Scale bar = 200 µm.

The online version of this article includes the following figure supplement(s) for figure 1:

**Figure supplement 1.** Wnt-dependent *fzd7* expression and differentiation of the pericardium.

heart region (*Figure 1D–E*, Dex (+)). Similarly, Wnt signal activation (Dex treatment) increased expression of the pericardium marker gene, *gata5*. Consistently, Wnt signal activation decreased that of a myocardium marker gene, *myosin light chain 2* (*myl2*). These changes indicate concomitant expansion or reduction of peri- and myocardium tissue, respectively. Wnt-dependent *fzd7* expression was also confirmed by targeted overexpression of Wnt6 using DNA injection (*Figure 1—figure supplement 1C-F*). As further confirmation, we treated embryos at the relevant stage with a Wnt signaling agonist, CHIR99021 (*Figure 1F*). We found *fzd7* expression expanding throughout cardiogenic mesoderm (*Figure 1G–H*). These results are also consistent with previous reports that overexpression of the Wnt antagonist, sFRP1, diminishes expression of *fzd7* in prospective pericardium, and *sfrp1* knockdown increases *fzd7* expression (*Gibb et al., 2013*). In addition, experimental knockdown of endogenous *wnt6* (by injection of antisense morpholino oligonucleotide (MO) for *wnt6* (*Lavery et al., 2008a*) at the one-cell stage) reduced *fzd7* expression around the heart region, and this was rescued by β-catenin activation at the tailbud stage (*Figure 1—figure supplement 1G-I*). Taken together, these results show that *fzd7* expression in cardiac tissues depends on Wnt signaling.

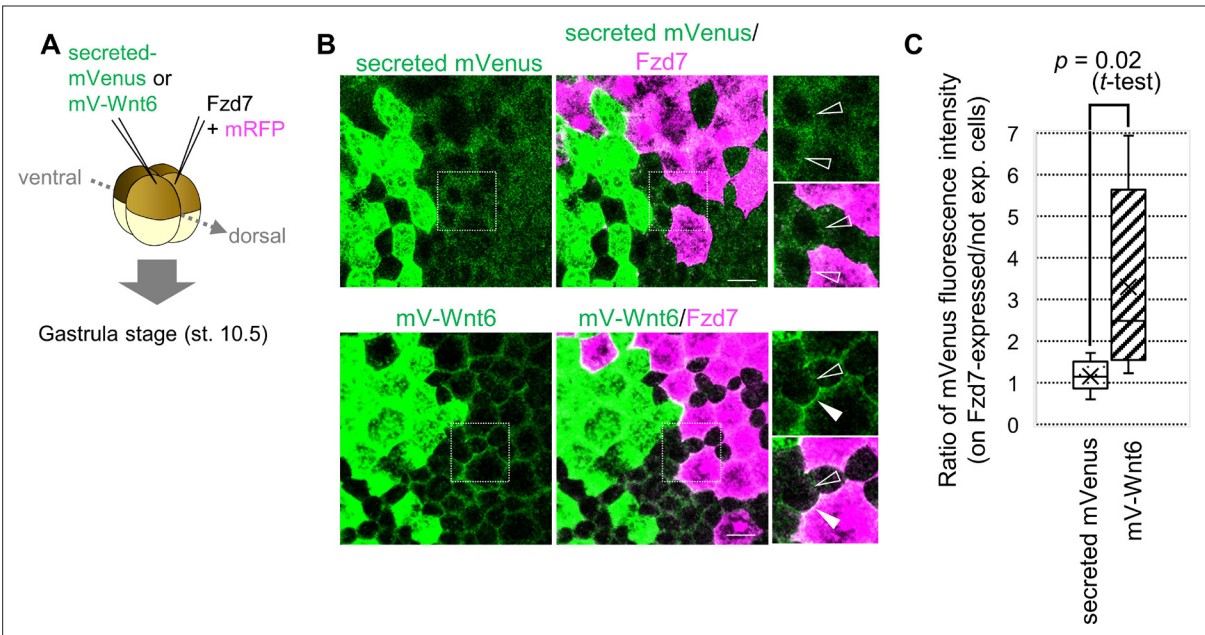

**Figure 2.** Wnt6 distribution is restricted by Fzd7. (**A**) Schematic view of the experiment. 500 pg of secreted mVenus (signal-peptide-fused mVenus) or mV-Wnt6 mRNA and Fzd7 mRNA (with mRFP) were injected into different blastomeres at the four-cell stage, and specimens were fixed at st.10.5 (gastrula stage). (**B**) mV-Wnt6 accumulated on Fzd7-expressing cells, whereas secreted mVenus did not (confocal image). mV-Wnt6 (green) accumulated on Fzd7-overexpressed cells (magenta cells; arrowhead), but not on intact cells (open-arrowhead). The distribution of secreted mVenus remained unchanged even when Fzd7 was overexpressed (open arrowheads). An enlarged view of the area indicated by squares on the left figure is shown on the right (the top, mVenus; the bottom, merged). Scale bar, 30 µm. (**C**) A box plot of the ratio of mVenus fluorescence intensity between neighboring cells (Fzd7-expressing cells / not Fzd7-overexpressed cells, n=7 pairs (secreted mVenus), 8 pairs (mV-Wnt6)). The horizontal line indicates the median. Edges of boxes indicate the first and third quartiles. The cross indicates the mean, and whiskers indicate the minimum and maximum.

The online version of this article includes the following figure supplement(s) for figure 2:

**Figure supplement 1.** Biological activity of mVenus-tagged Wnt6 and confirmation of the secretion of secreted mVenus.

## Fzd7 restricts the range of Wnt6 protein distribution

To examine whether Fzd7 restricts Wnt6 protein distribution, we used Wnt6 with an mVenus-tag. Functional activity of N-terminally tagged Wnt6 (mV-Wnt6) was nearly the same as that of intact Wnt6 (*Figure 2—figure supplement 1A, B*). At the four-cell stage, we injected mRNA of mV-Wnt6 (or secreted mVenus, as a negative control) and Fzd7 (with mRFP as a tracer) into two blastomeres (*Figure 2A*). mV-Wnt6 protein accumulated heavily on Fzd7-expressing cells, but secreted mVenus did not (*Figure 2B and C*). Next, we confirmed that the 'secreted' mVenus we used in this study actually was secreted. Fluorescence of 'secreted' mVenus was clearly detected on cells expressing anti-GFP antibody (morphotrap), far from the source of 'secreted' mVenus (*Figure 2—figure supplement 1C*). These results indicate that Fzd7 specifically accumulates Wnt6 on cell membranes.

## Exploring the functional significance of feedback regulation with computer simulation

To explore the potential biological significance of Wnt-dependent expression of *fzd7* in heart development, we examined it with computer simulation. As illustrated in *Figure 3—figure supplement 1A*, we assumed that (i) Wnt6 production is limited to the left region, 0–10 μm from the surface of the embryo (*Lavery et al., 2008a*); (ii) Fzd7 is broadly expressed in the middle region (at 10–40 μm) as measured in *Xenopus* embryos (*Figure 1—figure supplement 1B*); (iii) the concentration of molecules follows reaction-diffusion equations. Explicitly distinguishing the binding of the ligand to the receptor and subsequent signal activation (*Figure 1—figure supplement 1A*), we assumed that (iv) Wnt-signaling activity is proportional to the integral of the endocytosis rate of Wnt6-receptor complexes. In addition, we set two types of receptor expression separately: the initial concentration of receptor at the onset of simulation (hereafter 'initial expression') and the rate of receptor production in response to Wnt signaling (hereafter 'feedback expression'; see Materials and methods for details).

The Wnt signaling gradient became steeper with increased feedback strength (*Figure 3A*). Next, we measured the width of the Wnt-signal-active region, setting a threshold of signal activation (*Figure 3A*, dashed line). The Wnt-signal-active region was restricted in the presence of the feedback. Here, feedback regulation appears sufficient for activation of Wnt signaling in a well-defined, narrow band. However, we wondered if such narrow activation can be achieved without this feedback regulation, but instead with high initial expression of the receptor, which also inhibits ligand diffusion. To examine this possibility, we assumed various initial amounts of the receptor. Higher initial expression resulted in a steep gradient (*Figure 3—figure supplement 1B*; for instance, see panel of 'feedback = zero, initial amount = 4 times'). Thus, narrow-band Wnt-signal activation can theoretically be achieved with either condition. However, higher initial expression to produce high levels of Fzd protein throughout cardiogenic mesoderm is biologically costly.

Developmental patterning is generally challenged by variations of ligand production. To further examine the importance of feedback, we focused on variability of ligand production. It has been suggested that ligand-dependent expression of the receptor makes the system robust against ligand production variation (*Eldar et al., 2003*). In *Eldar et al., 2003*, signal activation was simulated only in a steady state. However, in most *in vivo* situations, it is not always possible to achieve a steady state. Extracellular factors such as HS proteoglycan (HSPG) delay the time to reach a steady state (*Eldar et al., 2003*). Thus, we simulated a system not limited to a steady state. We examined the response of a system with a 50% increase in ligand production as an extreme case of gene expression change (*Figure 3—figure supplement 2A*). We calculated differences of the positions of the edge of the Wnt-signal-active region between the two Wnt production rates (*Figure 3—figure supplement 2B*). The differences were calculated for all possible thresholds of Wnt signal activation. We found that differences in edge positions were small with feedback and/or a high initial amount of receptor. These numerical simulations suggested that a system with feedback regulation or a high initial amount of receptor is robust against variations in ligand production.

In the above simulation results, signal levels were shown at a certain time point (~1 day after the onset of simulation), which corresponds to the time for heart development in *Xenopus*. However, developmental systems are also often challenged by variations in the speed of developmental events. To examine the effects of this variability, we next visualized the time course of activation level and molecule concentrations with a heat map. Briefly, by showing Wnt signaling activity at each time point with a color scale, line graphs were converted to heatmap rows, and these were combined along

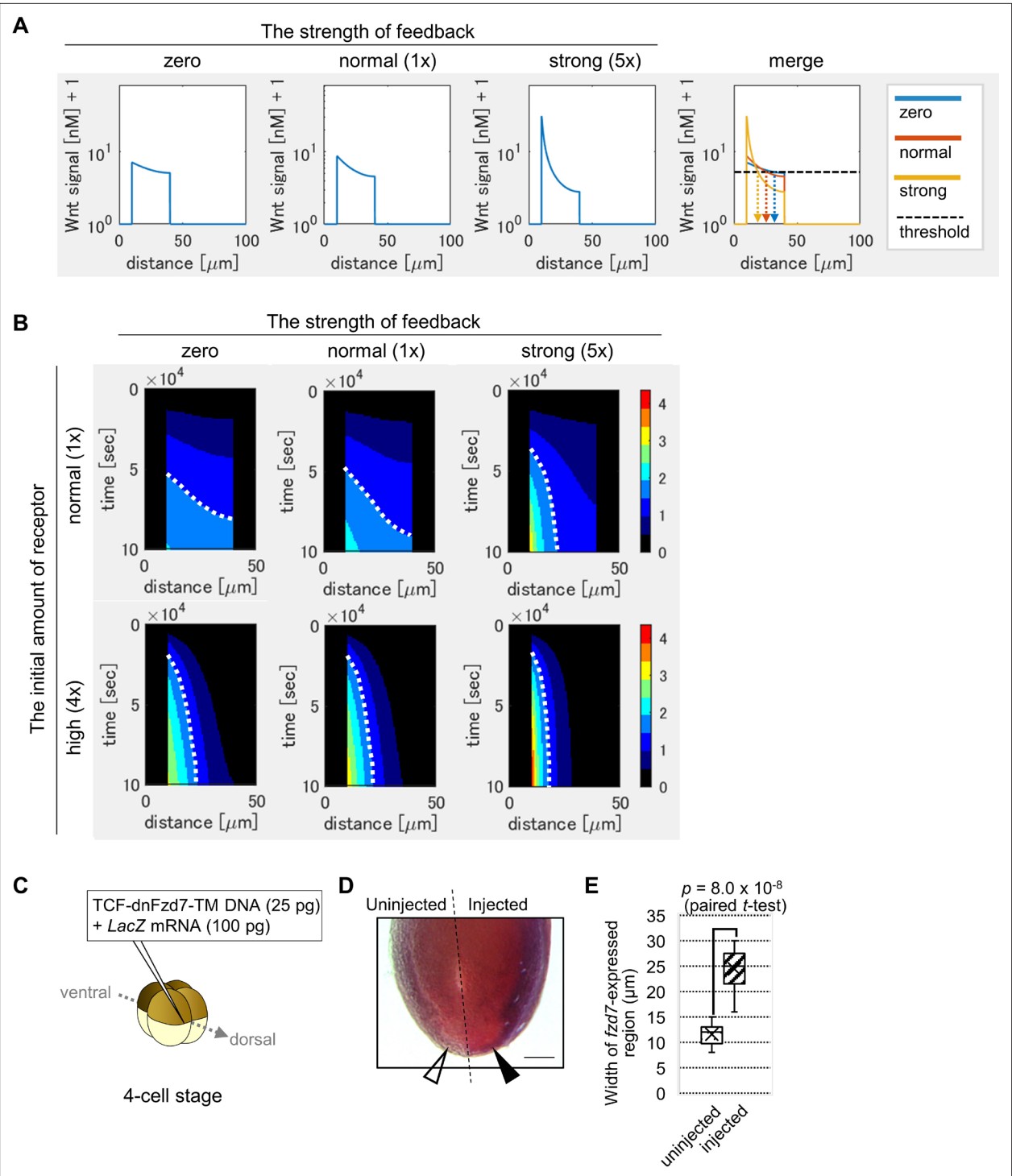

**Figure 3.** Feedback regulation of *fzd7* expression restricts the Wnt signal active region. (**A**) Wnt-dependent expression of *fzd7* increased Wnt signal activity near the Wnt source and decreased Wnt signal activity far from the source, making a steep gradient. An example of the threshold of Wnt signal activity to determine the edge of Wnt-signal-active region is shown by a dashed line (merge). Arrows show the edge of the activated region in each condition. 'strong' means 5-times higher strength of feedback than 'normal'. These figures correspond to a column in *Figure 3—figure supplement 1B* ('the initial amount of receptor = normal'). (**B**) Heatmap of Wnt signaling activity. The y-axis represents time elapsed from top to bottom (*Figure 3— figure supplement 3* for the detail). The dotted line shows an example of temporal changes of the edge of the Wnt-signal-active region. The line became nearly vertical under strong feedback with a 'normal' initial amount of receptor, indicating that a system with strong feedback quickly reaches a steady state. In the presence of high initial expression (4-times higher than the 'normal'), the slope of contour was nearly perpendicular, even without feedback. These figures correspond to a column in *Figure 3—figure supplement 4A* ('the initial amount of receptor' = 'normal' or '4 times'). (**C**) TCF-

*Figure 3 continued on next page*

*Figure 3 continued*

dnFzd7-TM plasmid (25 pg) was injected into the marginal region of a dorsal blastomere at the four-cell stage with a tracer, *LacZ* mRNA (100 pg). (**D**) *In situ* hybridization using a *fzd7* intracellular domain probe. The probe recognizes the intracellular domain of *fzd7*, which the dominant negative *fzd7* lacks. Thus, it visualizes endogenous expression of full-length *fzd7* (pericardium region), or the Wnt signal active region. The *fzd7* expression area, or Wnt signal active region, was expanded (arrowhead) by injection of TCF-dnFzd7-TM plasmids, as quantified in E. (**E**) A box plot of the width of *fzd7*-expressing region in the uninjected side (open arrowhead) and the injected side (arrowhead). The horizontal line indicates the median. Edges of boxes indicate the first and third quartiles. The cross indicates the mean, and whiskers indicate the minimum and maximum (n=14 each).

The online version of this article includes the following figure supplement(s) for figure 3:

**Figure supplement 1.** Mathematical modeling of Wnt signaling activity in the heart region.

**Figure supplement 2.** Changes in Wnt signaling activity due to varying Wnt production.

**Figure supplement 3.** Method of heatmap drawing.

**Figure supplement 4.** Time changes of Wnt signaling activity without sFRP1 (**A**) or with sFRP1 (**B**).

**Figure supplement 5.** Time changes of free ligand amount without sFRP1 (**A**) or with sFRP1 (**B**).

**Figure supplement 6.** Time changes of free receptor amount without sFRP1 (**A**) or with sFRP1 (**B**).

**Figure supplement 7.** TCF-dnFzd7-TM accumulates Wnt6 protein but cell-autonomously inhibits Wnt signaling.

**Figure supplement 8.** dnFzd7 with only 'trap-Wnt' function does not expand Wnt-signal-active region but dnFzd7 with only 'inactivate-Fzd7' function does.

the axis of time (*Figure 3—figure supplement 3*). The dotted curve aligned between two colors, which shows the contour of activation level, was almost vertical in the case of 'strong' feedback with a 'normal' initial amount of receptor (*Figure 3B*, 'strong' feedback is five times stronger than 'normal' feedback). This means that with strong feedback, the boundary position of differentiation of tissues did not change with time. In other words, a system with feedback quickly reaches a steady state of the boundary position. In contrast, with no feedback, the contour was curved, indicating slow progress toward a steady state (*Figure 3B*). The effect of feedback was evident with lower initial amounts of receptor (*Figure 3—figure supplement 4A*). Contrarily, in the presence of high initial expression (four times higher than the 'normal' initial amount of receptor), the slope of contour was nearly perpendicular, even without feedback (*Figure 3B*). These simulations indicate that the feedback contributes to robustness not only against changes in Wnt secretion level, but also against changes in the developmental time window, especially when the initial amount of receptor is low.

## The role of receptor feedback *in vivo*

We next examined *in vivo* whether Wnt-dependent expression of *fzd7* is required for Wnt signaling activation. To test this, we planned to inhibit *fzd7* expression only after feedback initiation, apart from the initial expression. One way to achieve this would be to delete enhancers of the *fzd7* gene. However, such a feedback enhancer has not been identified so far. As another strategy, we wondered whether expression of a dominant-negative Fzd7 receptor in a Wnt-dependent manner could specifically interfere with *fzd7* expression, only after feedback was initiated. Thus, we synthesized *fzd7* lacking the intracellular domain required for Wnt signaling (that functions as a dominant-negative [*Abu-Elmagd et al., 2006*]) fused with a transmembrane sequence (to express it cell-autonomously). The dominant-negative Fzd7 (dnFzd7) is expressed in a Wnt-dependent manner from a modified TOPFLASH plasmid (*Figure 3—figure supplement 7A*). We expected that this construct would cell-autonomously inhibit Wnt signal transduction and subsequent *fzd7* expression in a Wnt-dependent manner. We confirmed that injection of this construct increases Wnt ligand localization to the cell membrane (*Figure 3—figure supplement 7B*) and inhibits intracellular Wnt signaling activity (*Figure 3—figure supplement 7C*). When the plasmid was injected into the prospective heart region (*Figure 3C*), interestingly, it expanded expression of pericardium markers, *fzd7* and *gata5* into the prospective myocardium region (*Figure 3D–E*, *Figure 3—figure supplement 7D*). Consistent with this, there was a corresponding restriction of expression of a myocardium marker, *tnni3* (*cardiac troponin I*) to a smaller region (*Figure 3—figure supplement 7D*). These results suggest that Wnt-dependent expression of dnFzd7 resulted in expansion of the Wnt-signal-active region.

dnFzd7 is expected to interfere with Wnt ligand clearance in two ways: 1. by trapping and sequestering Wnt on the cell membrane, and 2. by reducing the amount of functional Fzd7 via nonfunctional-heterodimer formation with Fzd7. This second way is expected because Fzd7 is considered to function

as a dimer (*Hirai et al., 2019*). Some may wonder whether competitive inhibition of Wnt clearance by trapping Wnt, instead of direct inhibition of Fzd7 caused expansion of the Wnt-signal-active region by dnFzd7. However, this was not the case. At the same time that dnFzd7 inhibits clearance, dnFzd7 also inhibits Wnt diffusion (*Figure 3—figure supplement 8A*). These two types of inhibition happen simultaneously and for the same duration. Thus, expansion of the Wnt-signal-active region by dnFzd7 is not caused by competitive inhibition of Wnt clearance. To further confirm this, we separately simulated the two functions of dnFzd7: competitive inhibition via Wnt protein binding (the 'trap-Wnt' function) and direct inhibition via heterodimer formation with Fzd7 (the 'inactivate-Fzd7' function). dnFzd7 with only the 'trap-Wnt' function reduced signaling level over the entire cardiogenic mesoderm (*Figure 3—figure supplement 8B*). This confirmed that competitive inhibition of Wnt clearance does not result in promotion of Wnt diffusion. In contrast, dnFzd7 with only the 'inactivate-Fzd7' function, which directly inhibits clearance of Wnt, reduced the signaling level near the Wnt source, while increasing the signaling level far from the source (*Figure 3—figure supplement 8B*). Thus, expansion of the Wnt-signal-active region by dnFzd7 expression *in vivo* is likely caused by direct inhibition of Fzd7. To summarize, results of the experiment above indicate that receptors that exist after the feedback loop started to work, restrict the range of Wnt signaling activation.

## sFRP1 expands the range of Wnt6 distribution

Wnt6 and Fzd7 feedback regulation operates in a wider context of regulation by sFRP1 and probable regulation by heparan sulfate. sFRP1 is an antagonist of Wnt ligand, secreted mainly from prospective myocardium (*Figure 1A*), and is essential for normal differentiation of myocardium (*Gibb et al., 2013*). Therefore, we initially expected that sFRP1 would inhibit Wnt6 signaling. However, in early *Xenopus* development, a related protein, Frzb (sFRP3), expands the range of Wnt8 distribution (*Mii and Taira, 2009*). So, we also examined whether sFRP1 can expand Wnt6 protein distribution. To analyze distributions of both Wnt6 and sFRP1, we used mVenus-tagged proteins and found that Wnt6 seems to have a narrow distributional range, staying relatively close to Wnt6-secreting cells, whereas the distributional range of sFRP1 seems to be broad (*Figure 4—figure supplement 1A*).

We next analyzed the relationship of sFRP1 and Wnt6. The distribution of mVenus-Wnt6 was expanded by sFRP1 (*Figure 4—figure supplement 1A*). In this experiment, Wnt6 and sFRP1 were co-injected into a single blastomere. However, *in vivo*, the sources of these two secreted molecules are separate. So, we next injected them into different blastomeres, and confirmed that the range of mVenus-Wnt6 was wider when sFRP1 was injected (*Figure 4—figure supplement 1B*). These results indicate that the antagonist, sFRP1, expands the Wnt6 ligand distribution, possibly preventing its binding to the Fzd7 receptor, as in the case of Wnt8-Frzb functional interaction (*Mii et al., 2017*).

## Involvement of heparan sulfate in heart development

In general, morphogen distribution is thought to be regulated by binding to heparan sulfate (*Yan and Lin, 2009*). Two types of heparan sulfate modification, N-acetyl and N-sulfo, regulate Wnt8-mediated signaling (*Mii et al., 2017*). To examine involvement of these two types of heparan sulfate in Wnt6-mediated signaling, we utilized the enzyme, Ndst1, which modifies sugar chains to convert N-acetyl HS to N-sulfo HS. Wnt6 distribution was not substantially changed by Ndst1 expression (*Figure 4—figure supplement 1C*). In contrast, sFRP1 distribution was changed, and was not detected on Ndst1-expressing cells, suggesting that sFRP1 preferentially localizes to N-acetyl HS (*Figure 4A*). Consistently, sFRP1 was preferentially localized to *ndst1*-knocked-down cells (*Figure 4—figure supplement 1D*). To further confirm this, we visualized N-acetyl-rich HS and N-sulfo-rich HS with immunohistochemistry (IHC). sFRP1 was highly co-localized with N-acetyl-rich HS (*Figure 4B–D*), but not with N-sulfo-rich HS (*Figure 4D*; *Figure 4—figure supplement 1E*). In the heart, N-acetyl-rich HS was present in prospective pericardium, while N-sulfo-rich HS was not clearly detected (*Figure 4E*). Consistent with this, endogenous *ndst1* expression was not clearly detected in the heart region (data not shown).

To examine the importance of N-acetyl HS in heart development, we made a heat-inducible Ndst1 construct (hsp::Ndst1), modified from the hsp::EGFP plasmid (*Michiue and Asashima, 2005*; *Figure 4—figure supplement 2A*). We confirmed that when the hsp::Ndst1 plasmid was injected and heat shock was applied, N-acetyl HS decreased cell-autonomously (*Figure 4—figure supplement 2B*). Then, we injected this plasmid into the prospective heart region at the four-cell stage, and heat shock was applied at the tailbud stage (st. 24). In these embryos, the area of the myocardium region

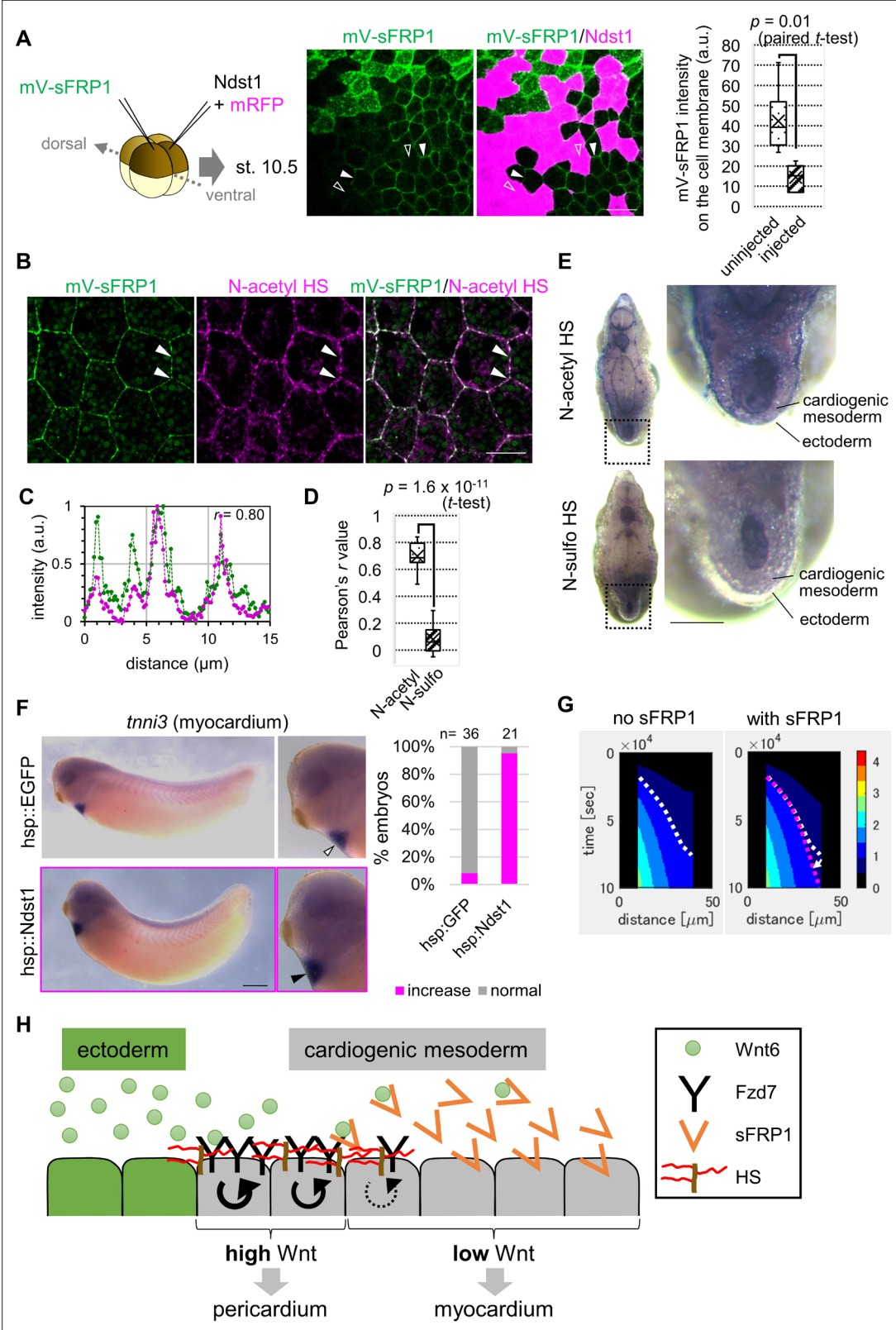

**Figure 4.** Contributions of sFRP1 and heparan sulfate in *Xenopus* heart development. (**A**) sFRP1 did not accumulate on *ndst1*-expressing cells (open arrowhead), compared to intact cells (arrowhead). Indicated mRNAs were injected into different blastomeres at the four-cell stage. *ndst1* was expressed in magenta-colored cells. mVenus fluorescence intensity on cell membranes is shown in the box plot. The horizontal line indicates the median. Edges of boxes indicate the first and third quartiles. The cross indicates the mean, and whiskers indicate the minimum and maximum. Scale bar, 30 μm. (**B**) sFRP1

*Figure 4 continued on next page*

*Figure 4 continued*

was highly localized on N-acetyl-rich HS. mV-sFRP1 mRNA was injected into one blastomere at the four-cell stage. N-acetyl-rich HS was visualized with IHC. Signal intensity is quantified in C-D. Scale bar, 20 μm. (**C**) A typical result of a signal intensity measurement (between the two arrowheads in B) (green, mV-sFRP1; magenta, N-acetyl-rich HS). (**D**) Quantification of the colocalization between sFRP1 and N-acetyl-rich HS or N-sulfo-rich HS. A box plot of Pearson's *r* value, calculated from measurements, as in C (n=10 (N-acetyl-rich HS), 11 (N-sulfo-rich HS)), The p-value from Student's *t*-test is shown. (**E**) The distribution of N-acetyl-rich and N-sulfo-rich HS in heart tissue. IHC for N-acetyl-rich and N-sulfo-rich HS. The dotted line in the left figure shows the position of the magnified image on the right. N-acetyl-rich HS was highly localized in the outer region of cardiogenic mesoderm (prospective pericardium). Scale bar, 0.1 mm. (**F**) Heat-induced expression of *ndst1* expanded *tnni3* expression (myocardium). hsp::Ndst1 or hsp::EGFP (***Figure 4— figure supplement 2A***) was injected into the prospective heart region at the four-cell stage, and heat shock (33°C, 15 min x 3 times) was induced at st. 24. *In situ* hybridization against *tnni3* (*cardiac troponin I*) was performed at st. 32 as quantified in the graph (Fisher's exact test, $p=2.8 \times 10^{-11}$). Scale bar, 0.5 mm. (**G**) Heatmap of Wnt signaling activity with/without sFRP1. The time change is shown as kymographs, in which time advances from top to bottom. The dotted line (white, no sFRP1; magenta, with sFRP1) shows an example of the threshold in Wnt signaling activity. The line became nearly vertical faster with sFRP1 (the change of the contour by addition of sFRP1 is indicated by a white arrow). These figures correspond to parts of ***Figure 3— figure supplement 4*** ('the initial amount of receptor' = 'normal', 'the strength of feedback' = 'normal'). (**H**) Schematic view of Wnt signaling in heart tissue. Wnt ligand is secreted from the epidermis (left cells). Before the onset of Wnt6 expression, the receptor (Fzd7) is broadly expressed around the prospective pericardium region. Wnt signaling is activated in a concentration-dependent manner (circular arrows), which induces Fzd7 expression and restricts Wnt ligand spreading. sFRP1 is secreted from the prospective myocardium region. This sFRP1 expression is inhibited by Wnt signaling. sFRP1 protein distribution is restricted by N-acetyl HS, which is abundant in the outer region of cardiogenic mesoderm.

The online version of this article includes the following figure supplement(s) for figure 4:

**Figure supplement 1.** Distribution and function of sFRP1.

**Figure supplement 2.** Heat-induced expression of *ndst1* reduces N-acetyl HS and restricts *gata5* expression.

was increased (***Figure 4F***) and that of the pericardium region was slightly decreased (***Figure 4—figure supplement 2C***). These results suggest that sFRP1 is trapped by N-acetyl-rich HS, inhibiting Wnt signaling just around the prospective myocardium region.

## Exploring roles of sFRP1 and heparan sulfate with computer simulation

To further examine contributions of sFRP1 and N-acetyl HS, we performed computer simulation. We assumed that (i) sFRP1 is expressed in the inner region of cardiogenic mesoderm (25–100 μm region from the surface of the embryo) (***Figure 3—figure supplement 1A***; ***Gibb et al., 2013***), (ii) sFRP1 expression is inhibited by Wnt signaling (***Gibb et al., 2013***), and (iii) N-acetyl HS is distributed in the prospective pericardium region (10–40 μm region in ***Figure 3—figure supplement 1A***), and N-sulfo HS is absent in the prospective heart region as in ***Figure 4E***.

The Wnt-signal-active region became narrow in the presence of sFRP1 and N-acetyl HS (***Figure 4G***). This tendency was clear when the initial expression level of the receptor and/or the strength of feedback was lower (***Figure 3—figure supplement 4B***). This may be because binding of Wnt to the receptor and that to sFRP1 compete. Most Wnt ligand was trapped by the receptor near the ligand source when the receptor amount is higher (***Figure 3—figure supplement 5A***), whereas Wnt spreads further and is broadly bound and inhibited by sFRP1 when the receptor amount is smaller. This makes the Wnt-signal gradient steeper. In addition, we compared the contour of Wnt signaling activity in the heatmap between conditions with and without sFRP1 (***Figure 4G***). In the presence of sFRP1, the contour became almost vertical faster, indicating that the position of a certain level of Wnt activity became steady faster than in the absence of sFRP1. Taken together, we suggest that the system with sFRP1 and N-acetyl HS reaches a steady state quickly. Feedback expression of *fzd7*, sFRP1, and N-acetyl HS imparts robustness against changes in the speed of development.

## Discussion

In this study, we analyzed patterning of cardiogenic mesoderm to specify a thin pericardium, in which endogenous Wnt signaling is highly activated. We found that the Wnt receptor, Fzd7, is expressed in a Wnt-dependent manner, and that this feedback facilitates robust formation of a steep gradient of Wnt signal activation. In addition, sFRP1 preferentially localized on N-acetyl-rich HS, which was abundant in prospective pericardium. This implies that N-acetyl HS traps sFRP1, thereby preventing sFRP1 from expanding much further into prospective pericardium. Thus, sFRP1 and HS contribute to shaping a steep gradient of Wnt activation and keeping the future pericardium thin (***Figure 4H***), leading to proper functioning of the heart.

To explore the functional significance of feedback regulation, we used computer simulation. We found there was another theoretically feasible way to restrict a highly activated region in morphogen-mediated signaling: by having a high amount of receptor initially, even without feedback regulation (*Figure 3—figure supplement 1B*). In this case, a high amount of receptor expression would need to be tightly regulated by an additional mechanism. However, considering the costs needed to produce high levels of Fzd protein throughout cardiogenic mesoderm, this strategy may not be biologically efficient. Starting with low expression everywhere and then specifically producing higher levels where necessary, via the discovered endogenous feedback mechanisms, may achieve higher fitness in nature. This idea is supported by a combination of two *in vivo* experiments as follows. First, sFRP1 expression, which represses only feedback expression, but maintains the initial expression, caused disappearance of *in situ* signal of *fzd7* (*Gibb et al., 2013*). This indicates that the initial *fzd7* expression level is below the threshold of *in situ* detection. Next, dnFzd7 expression, which spreads Wnt ligand, caused increased *fzd7* expression via feedback above the detection threshold. This suggests that Wnt accumulated near the source in absence of dnFzd7 was capable of inducing feedback *fzd7* expression above the detection threshold. Taken together, these results indicate that the feedback expression level is higher than the initial expression level. This way, dominance of the feedback strategy over the high initial expression strategy is experimentally validated.

Since ~83% of genes are differentially expressed in humans (*Storey et al., 2007*), the production of Wnt ligand is also expected to vary. Against variation in Wnt production, the system with feedback is robust and achieves reproducible patterning. Therefore, the feedback we discovered is essential to ensure efficient and reliable patterning. There are many factors that may perturb the reproducibility of morphogen-mediated signaling gradients *in vivo*. For instance, speed of morphogenesis can vary, for example, temperature-dependence in poikilotherms such as *Xenopus* (*Khokha et al., 2002*). Therefore, the amount of Wnt ligand, and the length of time for heart differentiation may vary among individuals in nature. At least considering these two types of variation, the discovered feedback regulation ensures reproducible patterning.

We showed that sFRP1 expands the Wnt protein distributional range (*Figure 4—figure supplement 1A*). However, distribution of ligand sometimes differs from that of signal activity because the protein does not always activate signaling, for example, when the protein is sequestered by an antagonist. Actually, the Wnt-signal-active region was restricted by sFRP1 in the simulation (*Figure 4G*).

Inhibition of Wnt signaling by sFRP1 results in some inhibition of feedback expression of *fzd7* and leads to further inhibition of Wnt signaling near the sFRP1 source. This steepens the Wnt signal activity gradient together with *wnt6-fzd7* feedback. However, there is a difference between the *wnt6-fzd7* feedback and sFRP1 mechanisms. Receptor-feedback can overcome variation of Wnt production, as discussed above. However, sFRP1 cannot buffer the variation of Wnt production, because sFPR1 expression is inhibited by Wnt, forming a positive feedback loop for Wnt signaling (*Gibb et al., 2013*).

When we experimentally interfered with this feedback, we found expansion of the pericardium region *in vivo* (*Figure 3D–E*), suggesting that *fzd7* in existence after feedback is initiated is necessary for normal patterning. Knockdown of sFRP1 also expands the pericardium region (*Gibb et al., 2013*). These observations suggest that feedback of *fzd7* expression alone is not sufficient, but is necessary together with sFRP1 to make the pericardium thin. In addition, computational modeling revealed that the time to achieve steady state positioning of the tissue boundary was shorter with sFRP1 and heparan sulfate (*Figure 4G*). To summarize, these results suggest that both regulatory mechanisms, *fzd7* feedback and sFRP1 with N-acetyl HS, ensure reproducible, thin patterning of the pericardium.

There are two gene regulatory circuits that can establish a robust system against variations of morphogen production (*Alon, 2006*). One is self-enhanced degradation via repression of receptor expression, in which the ligand inhibits receptor expression, and the receptor inhibits ligand degradation. This occurs in the *Drosophila* Wingless/Fzd pathway (*Cadigan et al., 1998*). The other is self-enhanced degradation via repression of receptor expression, in which the ligand induces receptor expression, and the receptor enhances ligand degradation. A well-known pathway of this type is the Hedgehog/Patched (Hh/Ptc) pathway (*Eldar et al., 2003*; *Goodrich et al., 1996*). Here we show that Wnt6/Fzd7 is another example of this type. The Hh/Ptc and Wnt6/Fzd7 systems are slightly different. Free Ptc without Hh acts as a negative regulator (*Taipale et al., 2002*). This additional negative feedback loop may attenuate the positive feedback of Hh/Ptc, leading to a milder effect of positive feedback. Thus, a gradient of Hh signaling activity may not be as steep as that of Wnt. For

the same reason, robustness of tissue boundary position may be higher in Wnt6 patterning than in Hh patterning. In other words, the network motif in Wnt6/Fzd7 circuit may be specialized to make extremely robust, steep morphogen gradients to pattern thin tissue.

We revealed that a modification of heparan sulfate, N-acetyl HS, is present in the outer region of cardiogenic mesoderm. N-acetyl modification of HS had previously been considered as just a precursor of other modifications, but we found sFRP1 to be one of specific binding partners of N-acetyl HS, in addition to Frzb (*Mii et al., 2017*) and other proteins related to morphogens (Yamamoto et al., in preparation). These results imply that differences in preference of N-acetyl HS binding may determine distributional and signaling ranges of ligands in general.

Wnt6 and sFRP1 molecules not only regulate normal embryonic heart development, but also regulate repair and regeneration after heart muscle injury in animal models of heart attack (myocardial infarction) (*Barandon et al., 2003*; *Schmeckpeper et al., 2015*). Our findings will be relevant to medical applications, for example, for drug design, since cell-surface molecules such as Frizzled or a specific modification of heparan sulfate or even the secreted molecule sFRP1, generally provide better drug targets than molecules inside cells. To reveal the precise regulation of morphogens and to consider medical applications, regulatory mechanisms of these components must be investigated further.

## Materials and methods
### *Xenopus* embryo manipulation and microinjection

All animal experiments were approved by the Office for Life Science Research Ethics and Safety, the University of Tokyo. For experiments conducted at the University of Aberdeen, all animal experiments were carried out under license from the United Kingdom Home Office (PPL PA66BEC8D). Manipulation of *Xenopus* embryos and microinjection experiments were carried out according to standard methods as previously described (*Sive et al., 2000*). Briefly, unfertilized eggs were obtained from female frogs injected with gonadotropin, and artificially fertilized with testis homogenate. Fertilized eggs were dejellied with 2% L-cysteine-HCl solution (pH7.8), and incubated in 1/10 x Steinberg's solution at 14–20°C. Embryos were staged according to *Nieuwkoop and Faber, 1967*. The amounts of injected mRNAs are described in the figure legends. For the experiments with the Wnt/β-catenin signaling agonist CHIR-99021, embryos were left to develop in 0.1×Steinberg's solution to embryonic stage 26/27, and transferred into the 5 µM solution (or DMSO as a control). For heat-shock induction, embryos were transferred into 33°C solution for 15 min, and then transferred into 16°C solution for 15 min. These steps were repeated three times as previously reported (*Michiue and Asashima, 2005*).

### Plasmid and RNA construction

The insertion sequence for the plasmids of monomeric Venus (mVenus/mV) fused with Wnt6 or sFRP1 were subcloned from pCS2 +xWnt6 (*Lavery et al., 2008b*) or pXT7-xFrzA (a gift from Dr. Sergei Sokol) (*Xu et al., 1998*), respectively. The PCR products for N-terminus tagged construct were inserted into pCSf107SP-mV-mT, and those for a C-terminus tag were inserted into pCSf107-mcs-mV-mT. An inducible β-catenin (Glucocorticoid Receptor (GR)-fused β-catenin) was used as previously described (*Afouda et al., 2008*). A *fzd7* fragment that was cut out from pCS2 +xFz7 ASN (a gift from Dr. Masanori Taira) was inserted into pCS2 +mcs-6MT-T vector using *Eco*RI/*Age*I sites. *gata5* was cloned from *Xenopus* cDNA and the product was inserted into pGEM-T easy vector (Promega Corp.).

TCF-dnFrizzled7-transmembrane plasmid (TCF-dnFzd7-TM) was a modified plasmid from the TOPFlash system, which is a well-known Wnt reporter construct, expresses luciferase in a Wnt-dependent manner. We replaced the luciferase sequence by the extracellular domain of *fzd7* (dominant negative Fzd7, dnFzd7) with mouse IgG transmembrane sequence (TM) (*Figure 3—figure supplement 7A*). This construct expresses the extracellular domain of Fzd7 on the cell surface, dependent on Wnt signaling activity, which then cell-autonomously inhibits Wnt-dependent endogenous *fzd7* expression.

hsp::Ndst1 plasmid was modified from hsp::EGFP (*Michiue and Asashima, 2005*). The Ndst1 sequence was subcloned from pCSf107-Ndst1-T (that was also used for the template for mRNA and RNA probe synthesis) (*Mii et al., 2017*). The PCR product was inserted into the vector, *Not*I and *Mlu*I

treated hsp::EGFP-hsp::caMLC, whose caMLC sequence was enzymatically deleted (Kaneshima et al., in preparation; *Figure 4—figure supplement 2A*).

mRNAs were transcribed *in vitro* using mMessage mMachine SP6 kit (Ambion). All the primers used for cloning were listed in *Supplementary file 2*.

## qRT-PCR

Total RNA was isolated from whole embryos using the RNeasy Mini Kit, according to manufacturer's instructions (QIAGEN) for processing of animal tissues (see also *Lee-Liu et al., 2012*; *Nakamura et al., 2016*). The abundance of RNAs was determined using a LightCycler 480 and SYBR Green I Master Reagents (Roche). Relative expression levels of genes were determined using ΔΔCt.

## Whole mount *in situ* hybridization (WISH)

WISH was performed based on *Xenopus* standard methods (*Harland, 1991*) with slight modifications in the duration of washes and hybridization temperature of 65°C. Plasmids for RNA probe synthesis were linearized and transcribed *in vitro* using DIG RNA labeling mix (Roche). Enzymes for RNA probe synthesis were listed in *Supplementary file 1*.

## Immunohistochemistry

*Xenopus* gastrula embryos were fixed with MEMFA (0.1 M MOPS, pH 7.4, 2 mM EGTA, 1 mM MgSO4, 3.7% formaldehyde) and immunostained by standard protocols with Tris-buffered saline (*Sive et al., 2000*). The specimens were incubated overnight at 4°C with the following primary antibodies: anti-N-acetyl HS (NAH46 [*Suzuki et al., 2008*], in-house preparation, 1:50), anti-N-sulfo HS (HepSS-1 (*Kure and Yoshie, 1986*), in-house preparation, 1:1000), diluted with 2% BSA in TBT (0.01% TritonX-100 in TBS). Following this, the samples were incubated with the secondary antibodies overnight at 4°C: anti-rabbit or mouse Alexa 488 or 555 antibody (Invitrogen). The specimens were visualized by the confocal microscope (FV-1200, Olympus). The signal intensity was measured by Fiji software (ImageJ 1.53f51; Java 1.8.0_172 (64-bit)) (*Schindelin et al., 2012*).

## Mathematical model

We used one-dimensional reaction-diffusion equations to simulate Wnt-signal gradient formation. We assumed that two molecules of Fzd7 bind one molecule of Wnt, based on recent X-ray structural analysis (*Hirai et al., 2019*).

The following chemical reactions were assumed. Molecules are denoted as follows: Wnt6, W; sFRP1, S; N-acetyl HS, H; Fzd7, R; dnFzd7, D.

$$\text{W} + 2\text{R} \underset{k_{-1}}{\overset{k_1}{\rightleftharpoons}} \text{WR}_2 \xrightarrow{k_4} \text{Wnt signal} (\omega) \tag{C1}$$

$$\text{W} + \text{S} \underset{k_{-2}}{\overset{k_2}{\rightleftharpoons}} \text{WS} \tag{C2}$$

$$\text{W} + \text{SH} \underset{k_{-2}}{\overset{k_2}{\rightleftharpoons}} \text{WSH} \tag{C3}$$

$$\text{S} + \text{H} \underset{k_{-3}}{\overset{k_3}{\rightleftharpoons}} \text{SH} \tag{C4}$$

$$\text{WS} + \text{H} \underset{k_{-3}}{\overset{k_3}{\rightleftharpoons}} \text{WSH} \tag{C5}$$

$$\text{W} + 2\text{D} \underset{k_{-1}}{\overset{k_5}{\rightleftharpoons}} \text{WD}_2 \tag{C6}$$

$$W + R + D \xrightleftharpoons[k_{-1}]{k_6} WRD \tag{C7}$$

$$R + D \xrightleftharpoons[k_{-1}]{k_7 \cdot [W]} RD \tag{C8}$$

$$D + D \xrightleftharpoons[k_{-1}]{k_7 \cdot [W]} D_2 \tag{C9}$$

$$\varnothing \xrightarrow{p_1 \cdot f(x)} W \tag{C10}$$

$$\varnothing \xrightarrow{p_3 \cdot k_4 \cdot [WR_2]} R \tag{C11}$$

$$\varnothing \xrightarrow{p_4 \cdot k_4 \cdot [WR_2]} D \tag{C12}$$

$$\varnothing \xrightarrow{p_2 \cdot \frac{1}{\left(\frac{\omega}{K_d}\right)^n + 1} \cdot g(x)} S \tag{C13}$$

Dominant-negative Fzd7 (D) may have two types of functions: to inactivate Fzd7, which decreases activatable Fzd7 by forming heterodimers (RD) in reaction (C7) or (C8), and to trap Wnt, which traps Wnt in the form of homo/heterodimers ($D_2$ , RD) in reaction (C6) and/or (C7).

To distinguish the two functions in the simulation, we set parameters as follows. In condition (iii) in the following table, we selectively deleted the 'trap-Wnt' function by deleting reactions (C6) and (C7). We set the binding rate constant of Wnt (W) and dnFzd7-containing multimers (RD and $D_2$) (k5 and k6) to zero. Here, the association rate constant of Fzd7 (R) and dnFzd7 (D) was set to $k_1[W]$ so that the Wnt-dependent 'inactivate-Fzd7' function remains in the form of reaction (C8).

In condition (iv), we selectively deleted the 'inactivate-Fzd7' function, deleting reaction (C7) by setting the association rate constants of Fzd7 (R) and dnFzd7 (D) ($k_6$) to zero.

| Condition | $p_4$ | $k_5$ | $k_6$ | $k_7$ |
|---|---|---|---|---|
| (i) No dnFzd7 production | 0 | $k_1$ | $k_1$ | 0 |
| (ii) dnFzd7 with both of two functions | $100 \cdot p_3$ | $k_1$ | $k_1$ | 0 |
| (iii) dnFzd7 with only 'inactivate-Fzd7' function | $100 \cdot p_3$ | 0 | 0 | $k_1$ |
| (iv) dnFzd7 with only 'trap-Wnt' function | $100 \cdot p_3$ | $k_1$ | 0 | 0 |

The above chemical equations were converted into the following differential equations.

$$\frac{\partial [W]}{\partial t} = \underbrace{D \frac{\partial^2 [W]}{\partial x^2}}_{\text{diffusion}} \underbrace{-k_1[W][R]^2 + k_{-1}[WR_2] - k_2[W][S] + k_{-2}[WS] - k_2[W][SH] + k_{-2}[WSH]}_{\text{binding to other molecules}}$$

$$\underbrace{-k_5[W][D]^2 + k_{-1}[WD_2] - k_6[W][R][D] + k_{-1}[WRD]}_{\text{binding to other molecules}} + \underbrace{p_1 \cdot f(x)}_{\text{production}} \tag{D1}$$

$$\frac{\partial [S]}{\partial t} = \underbrace{D \frac{\partial^2 [S]}{\partial x^2}}_{\text{diffusion}} \underbrace{-k_2[W][S] + k_{-2}[WS] - k_3[S][H] + k_{-3}[SH]}_{\text{binding to other molecules}} + \underbrace{p_2 \cdot \frac{1}{(\frac{\omega}{K_d})^n + 1} \cdot g(x)}_{\text{production (inhibited by Wnt signal)}} \tag{D2}$$

$$\frac{\partial [WS]}{\partial t} = \underbrace{D \frac{\partial^2 [WS]}{\partial x^2}}_{\text{diffusion}} \underbrace{+k_2[W][S] - k_{-2}[WS] - k_3[WS][H] + k_{-3}[WSH]}_{\text{binding to other molecules}} \tag{D3}$$

$$\frac{\partial [H]}{\partial t} = \underbrace{-k_3[S][H] + k_{-3}[SH] - k_3[WS][H] + k_{-3}[WSH]}_{\text{binding to other molecules}} \tag{D4}$$

$$\frac{\partial[SH]}{\partial t} = \underbrace{-k_2[W][SH] + k_{-2}[WSH] + k_3[S][H] - k_{-3}[SH]}_{\text{binding to other molecules}} \tag{D5}$$

$$\frac{\partial[WSH]}{\partial t} = \underbrace{k_2[W][SH] - k_{-2}[WSH] + k_3[WS][H] - k_{-3}[WSH]}_{\text{binding to other molecules}} \tag{D6}$$

$$\frac{\partial[R]}{\partial t} = \underbrace{2(-k_1[W][R]^2 + k_{-1}[WR_2])}_{\text{binding to other molecules}}$$
$$\underbrace{-k_6[W][R][D] + k_{-1}[WRD] - k_7[W][R][D] + k_{-1}[RD]}_{\text{binding to other molecules}} \tag{D7}$$
$$\underbrace{+p_3 \cdot k_4[WR_2]}_{\text{production}}$$

$$\frac{\partial[WR_2]}{\partial t} = \underbrace{k_1[W][R]^2 - k_{-1}[WR_2]}_{\text{binding to other molecules}} \; \underbrace{-k_4[WR_2]}_{\text{degradation}} \tag{D8}$$

$$\frac{\partial[D]}{\partial t} = \underbrace{2(-k_5[W][D]^2) + k_1[WD_2]}_{\text{binding to other molecules}}$$
$$\underbrace{-k_6[W][R][D] + k_{-1}[WRD] - k_7[W][R][D] + k_{-1}[RD]}_{\text{binding to other molecules}} \tag{D9}$$
$$\underbrace{+2(-k_7[W][D]^2) + k_{-1}[D_2]}_{\text{binding to other molecules}}$$
$$\underbrace{+p_4 \cdot k_4[WR_2]}_{\text{production}}$$

$$\frac{\partial[RD]}{\partial t} = \underbrace{k_7[W][R][D] - k_{-1}[RD]}_{\text{binding to other molecules}} \tag{D10}$$

$$\frac{\partial[D_2]}{\partial t} = \underbrace{k_7[W][D]^2 - k_{-1}[D_2]}_{\text{binding to other molecules}} \tag{D11}$$

$$\frac{\partial[WD_2]}{\partial t} = \underbrace{k_5[W][D]^2 - k_{-1}[WD_2]}_{\text{binding to other molecules}} \tag{D12}$$

$$\frac{\partial[WRD]}{\partial t} = \underbrace{k_6[W][R][D] - k_{-1}[WRD]}_{\text{binding to other molecules}} \tag{D13}$$

$$\text{Wnt signaling activity} = k_4 \int_0^t [WR_2]\, dt \equiv \omega \tag{D14}$$

Wnt signaling activity is defined as the integral of the endocytosis rate of the Wnt6-receptor complex (*Equation D14*), resembling the transcriptional activator β-catenin, which accumulates in the nucleus to activate transcription of Wnt-responsive genes (*Logan and Nusse, 2004*; *MacDonald et al., 2009*). The time delay is not explicitly implemented, since Wnt signal transduction inside cells is considered to be quick, as transcription of the target gene is detected within 15 min (*Kafri et al., 2016*).

Production of Wnt6 or sFRP1 was limited to left and right regions, respectively.

$$f(x) = \begin{cases} 1 & (x \le 10\ \mu m) \\ 0 & (x > 10\ \mu m) \end{cases}$$

$$g(x) = \begin{cases} 0 & (x < 25\ \mu m) \\ 1 & (x \ge 25\ \mu m) \end{cases}$$

Initial concentrations of molecules are set to zero, except for that of the receptor ($R_0$) and N-acetyl HS ($H_0$). The receptor and N-acetyl HS are assumed to be present at 10 μm < x < 40 μm.

$$R_0 = \begin{cases} r_0[\sin\left(\frac{\pi(x-10)}{15} - \frac{\pi}{2}\right) + 1] & (10\,\mu m < x < 40\,\mu m) \\ 0 & (x \leq 10\,\mu m,\ x \geq 40\,\mu m) \end{cases}$$

$$H_0 = \begin{cases} h[\sin\left(\frac{\pi(x-10)}{15} - \frac{\pi}{2}\right) + 1] & (10\,\mu m < x < 40\,\mu m) \\ 0 & (x \leq 10\,\mu m,\ x \geq 40\,\mu m) \end{cases}$$

The simulated field represents a lateral half of the developing heart. Considering the bilateral symmetry of the heart, concentrations of Wnt6, sFRP1, and the Wnt6-sFRP1 complex at the ends of the field (x=0 μm, $x_{max}$) are assumed to be the same in both halves. Thus, the Neumann boundary condition is assumed (15).

$$\frac{\partial\,[W]\,,[S]\,,[WS]}{\partial x}\bigg|_{x=0\,\mu m,\ x_{max}} = 0 \tag{B1}$$

## Parameter values

| | |
|---|---|
| $k_1 = 2.66 \times 10^{-6}\ nM^{-2}second^{-1}$ | Association rate constant of Wnt6 and Fzd7, set in this study |
| $k_{-1} = 9.6 \times 10^{-5}\ second^{-1}$ | Dissociation rate constant of Wnt3a and Fzd8 (**Bourhis et al., 2010**) |
| $k_2 = 4.33 \times 10^{-5}\ nM^{-1}second^{-1}$ | Association rate constant of Wnt3a and sFRP1 (**Wawrzak et al., 2007**) |
| $k_{-2} = 4.86 \times 10^{-4}\ second^{-1}$ | Dissociation rate constant of Wnt3a and sFRP1 (**Wawrzak et al., 2007**) |
| $k_3 = 1.86 \times 10^{-4}\ nM^{-1}second^{-1}$ | Association rate constant of HSPG and Gremlin (**Chiodelli et al., 2011**) |
| $k_{-3} = 3.66 \times 10^{-3}\ second^{-1}$ | Dissociation rate constant of HSPG and Gremlin (**Chiodelli et al., 2011**) |
| $k_4 = 2.52 \times 10^{-4}\ second^{-1}$ | Internalization rate constant of Wnt-receptor complex, set in this study |
| $p_1 = 1 \times 10^{-4}\ nM\ second^{-1}$ | Normal production rate of Wnt, set in this study |
| $p_2 = 1 \times 10^{-3}\ nM\ second^{-1}$ | Production rate of sFRP1, set in this study |
| $p_3 = 10$ | Relative production rate of the receptor at normal setting of the feedback strength. This is the rate relative to that of Wnt-receptor complex internalization, set in this study |
| $K_d = 1 \times 10^{-2}\ nM$ | Dissociation constant in Hill equation of repression of sFRP1 expression by Wnt signaling activity, set in this study |
| $n = 2$ | Hill coefficient in Hill equation of repression of sFRP1 expression by Wnt signaling activity, set in this study |
| $D = 20\ \mu m^2 second^{-1}$ | Diffusion coefficient of Wnt8 (**Mii et al., 2021**) |
| $r_0 = 75\ nM$ | Normal concentration of the receptor at t=0, set in this study |
| $h = 1 \times 10^8\ nM$ | Concentration of heparan sulfate, set in this study |
| $x_{max} = 1 \times 10^2\ \mu m$ | Width of developmental field of heart in *Xenopus* |
| $t_{max} = 1 \times 10^5\ sec$ | The time length of heart development ($\simeq 1$ day) |

We non-dimensionalized the reaction diffusion equation with the following scaling factors:

| | |
|---|---|
| $t^* = t_{max} - t_{min}$ | scaling factor for non-dimentionalization of time |
| $x^* = \sqrt{4D\left(t_{max} - t_{min}\right)}$ | scaling factor for non-dimentionalization of distance |
| $c^* = p_1 \cdot \left(t_{max} - t_{min}\right)$ | scaling factor for non-dimentionalization of concentration |

## Nondimensional equations

Using these parameters, we non-dimensionalized parameters and variables in the reaction diffusion equations as follows:

$$\widetilde{k_{1,5,6,7}} = k_{1,5,6,7} \cdot c^{*2} \cdot t^*$$

$$\widetilde{k_{2,3}} = k_{2,3} \cdot c^* \cdot t^*$$

$$\widetilde{k_{-1,-2,-3,4}} = k_{-1,-2,-3,4} \cdot t^*$$

$$\widetilde{D} = D \cdot t^*/x^{*2}$$

$$\widetilde{p_{1,2}} = p_{1,2} \cdot t^*/c^*$$

$$\widetilde{x} = x/x^*$$

$$\widetilde{t} = t/t^*$$

$$\widetilde{[molecule]} = [molecule]/c^*$$

Numerical simulation was performed using the partial differential equation solver (pdepe) in MATLAB (MathWorks, version: R2020a) (**Source code 1**).

## Acknowledgements

We thank Dr. Yukio Nakamura (Aberdeen University, UK; Present address: Repertoire Genesis Inc, Japan.) and Dr. Masanori Taira (Chuo University, Japan) for their help in initiating this project; Dr. Mary Elizabeth Pownall (University of York, UK) for discussion; Dr. Takehiko Nakamura (Seikagaku Corporation, Japan) for NAH46 antibody and hybridoma; Dr. Osamu Yoshie (Kindai University, Japan) for HepSS-1 hybridoma; Dr. Makoto Matsuyama (Shigei Medical Research Institute, Japan) for the contribution to the generation of NAH46 and HepSS-1 antibody from the hybridomas; Dr. Steven D Aird for technical editing of the manuscript. This international collaboration was supported in part by Daiwa Anglo-Japanese Foundation (12969/13787 to TY, BA, TM, and SH); with additional research support in Japan from MEXT/JSPS KAKENHI (19K16138 to TY, 18K06244/21K06183 to TY and TM); and in the United Kingdom from BHF (RG/18/8/33673 to SH) and BBSRC (BB/N021924/1; BB/M001695/1 to SH). SH was a Royal Society/Leverhulme Trust Senior Research Fellow (SRF\R1\191017). We also thank National BioResource Project (NBRP) and NBRP Information Center (National Institute for Genetics) for providing us *Xenopus* genomic database (http://viewer.shigen.info/xenopus/).

# Additional information

## Funding

| Funder | Grant reference number | Author |
|---|---|---|
| Daiwa Anglo-Japanese Foundation | 12969/13787 | Takayoshi Yamamoto Boni A Afouda Tatsuo Michiue Stefan Hoppler |
| Ministry of Education, Culture, Sports, Science and Technology | 19K16138 | Takayoshi Yamamoto |
| Ministry of Education, Culture, Sports, Science and Technology | 18K06244 | Takayoshi Yamamoto Tatsuo Michiue |
| Ministry of Education, Culture, Sports, Science and Technology | 21K06183 | Takayoshi Yamamoto Tatsuo Michiue |
| British Heart Foundation | RG/18/8/33673 | Stefan Hoppler |
| Biotechnology and Biological Sciences Research Council | BB/N021924/1 | Stefan Hoppler |
| Biotechnology and Biological Sciences Research Council | BB/M001695/1 | Stefan Hoppler |
| Leverhulme Trust | SRF\R1\191017 | Stefan Hoppler |

The funders had no role in study design, data collection and interpretation, or the decision to submit the work for publication.

## Author contributions

Takayoshi Yamamoto, Conceptualization, Data curation, Formal analysis, Funding acquisition, Investigation, Methodology, Project administration, Resources, Software, Supervision, Validation, Visualization, Writing - original draft, Writing - review and editing; Yuta Kambayashi, Conceptualization, Data curation, Formal analysis, Investigation, Methodology, Resources, Validation, Visualization, Writing - original draft, Writing - review and editing; Yuta Otsuka, Conceptualization, Data curation, Formal analysis, Investigation, Methodology, Software, Validation, Visualization, Writing - original draft, Writing - review and editing; Boni A Afouda, Data curation, Funding acquisition, Investigation, Validation, Visualization; Claudiu Giuraniuc, Investigation; Tatsuo Michiue, Conceptualization, Funding acquisition, Resources, Supervision, Writing - review and editing; Stefan Hoppler, Conceptualization, Funding acquisition, Methodology, Project administration, Resources, Supervision, Writing - original draft, Writing - review and editing

## Author ORCIDs

Takayoshi Yamamoto (iD) http://orcid.org/0000-0001-8028-6050
Yuta Kambayashi (iD) http://orcid.org/0000-0001-5219-6592
Yuta Otsuka (iD) http://orcid.org/0000-0003-3872-4349
Claudiu Giuraniuc (iD) http://orcid.org/0000-0001-7127-2642
Tatsuo Michiue (iD) http://orcid.org/0000-0001-9047-0513
Stefan Hoppler (iD) http://orcid.org/0000-0003-0730-4798

## Ethics

All animal experiments were approved by The Office for Life Science Research Ethics and Safety, the University of Tokyo. For experiments conducted at the University of Aberdeen, all animal experiments were carried out under license from the United Kingdom Home Office (PPL PA66BEC8D).

## Decision letter and Author response

Decision letter https://doi.org/10.7554/eLife.73818.sa1
Author response https://doi.org/10.7554/eLife.73818.sa2

# Additional files

## Supplementary files

- Supplementary file 1. RNA probe synthesis for *in situ* hybridization.
- Supplementary file 2. PCR primers used in this study.
- Transparent reporting form
- Source code 1. Computer simulation code written in MATLAB. Running this code in MATLAB (version: R2020a) generates all the computer simulation results in this article. Figures are named using following indices for different parameter sets. 'indexWntproduction = 1, 2, 3' indicate 50% decrease, normal, and 50% increase of Wnt production, respectively. 'indexRinitialcondition = 1, 2, 3, 4, 5' indicate zero, half, normal, 2 times, and 4 times of 'the initial amount of receptor', respectively. 'indexRproduction = 1, 2, 3' indicate zero, normal, and 5 times of 'the strength of feedback', respectively. 'indexSFRPproduction = 1, 3' indicate 'without sFRP1', 'with sFRP1', respectively. 'indexDproduction = 1, 2' indicate 'without dnFzd7', 'with dnFzd7', respectively. 'indexDfunction = 1, 2, 3' indicates (1) dnFzd7 has normal dnFzd7 function, (2) has only 'inactivate-Fzd7' function, (3) has only 'trap-Wnt' function, respectively.

## Data availability

All data generated or analyzed during this study are included in the manuscript and supporting files. Mouse IgG transmembrane sequence have been deposited in Genbank/DDBJ under accession code V00776.

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
