## [Editor Report]

The study deals with the mechanisms that establish the Wnt gradient combining a mathematical model and experiments considering multiple extracellular components such as receptor and diffusible antagonist. The study revealed that the ligand/receptor feedback enables robust and quick formation of the morphogen gradient and that the diffusible antagonist also plays a role in this process.

---

## [Decision Letter]

**Decision letter after peer review:**

Thank you for submitting your article "Positive Feedback Regulation of *fzd7* Expression Robustly Shapes Wnt Signaling Range in Early Heart Development" for consideration by *eLife*. Your article has been reviewed by 3 peer reviewers, including Makoto Sato as the Reviewing Editor and Reviewer #1, and the evaluation has been overseen by Didier Stainier as the Senior Editor.

Essential revisions:

1) Addition of data quantification, model explanation and discussion of the results are needed.

2) The manuscript would greatly benefit from English language editing.

*Reviewer #1 (Recommendations for the authors):*

Addition of data quantification, model explanation and discussion of the results would significantly improve the paper.

*Reviewer #2 (Recommendations for the authors):*

Mathematical analysis

This type of mathematics is easy enough, and experimental biologists can do the analysis by themselves. Alternatively, if the authors cannot handle this, replace all the "mathematical" in the manuscript with "computational".

Citation

The reference the authors cited (Umulis et al., 2009) for determining reaction parameters is a review article (page 30). The authors should cite the original measurement (Hatta 2000 10.1002/1097-0282(2000)55:5<399::AID-BIP1014>3.0.CO;2-9, Iemura 1998 10.1073/pnas.95.16.9337.)

*Reviewer #3 (Recommendations for the authors):*

Abstract: Authors need to clearly distinguish between results from experiments and results from simulations.

For example, the reader cannot determine whether the following results are derived from experimental data or from simulations.

Lines 8-9, "We revealed…that this receptor-feedback is essential for shaping a steep gradient of Wnt".

Lines 9-10, "The feedback imparts robustness against fluctuations of Wnt ligand production and allows the system to reach a steady state quickly".

In addition, the result presented in the next sentence, "We found that…by restricting sFRP1 spreading" is quite ambiguous. The reader may mistakenly believe that all the data were obtained by experiments in the cardiac field.

Figure 1C, F It is quite difficult to determine whether the expression of fzd7 in the prospective heart file is affected or not. It would be better to add magnified images. Also, staining with gata5, a pericardial marker, can help identify this field in embryos.

Figure 2: Adding some negative control, for example, secreted mVenus (without Wnt), can increase the reliability of this result.

Supplemental Figure 6E: Analysis of the expression of marker genes in the pericardium and myocardium would improve the quality of this result.

Figure 4C, D: In Figure 4C, it is unclear whether N-acetyl HS is expressed in the prospective myocardium or not. Furthermore, the distribution of HS shown in Figure 4D is puzzling to me. Although the text explains that N-acetyl HS is expressed predominantly in the outer part of the cardiogenic mesoderm, in this figure HS is present in both the pericardium and myocardial regions. Probably this is due to an inadvertent mistake by the author, but I think this figure should be corrected.

[Editors' note: further revisions were suggested prior to acceptance, as described below.]

Thank you for resubmitting your work entitled "Positive feedback regulation of *fzd7* expression robustly shapes a steep Wnt gradient in early heart development, together with sFRP1 and heparan sulfate" for further consideration by *eLife*. Your revised article has been evaluated by Didier Stainier (Senior Editor) and a Reviewing Editor.

The manuscript was considerably improved compared with the previous version addressing the reviewer's comments. However, I feel that the conclusions authors are trying to make solely from the computational results are too strong.

The authors compare two alternative models that ensures Fzd7 expression that causes thin Wnt activation domain: Wnt/Fzd7 feedback and Wnt-independent Fzd7 expression. The authors argue that the results of computer simulations prefer the former model. However, I have an impression that both models are possible. The authors have to show a set of new computational results that clearly demonstrate that the former model is likely. Or the text should be modified so that both models are preferred according to the computational study. The results of *in vivo* experiment expressing TCF-dnFzd7 indicates that Wnt/Fzd7 feedback regulates Fzd7 expression. Based on this, possibility of Wnt-independent Fzd7 expression may be discussed. It's common in biology that multiple mechanisms cooperate to ensure robustness.

In Page 7, line 25-29, two alternative conditions 'no feedback and high expression' and 'with feedback and low expression' are compared. The magnitude of feedback and initial expression have to be indicated in the main text and figure. The Figure 3B legend says that the results are the same as Figure 3—figure supplement 2A. I don't think they are the same. Fix this point. The authors argue that the latter is more robust. But, what happens if the initial expression is further enhanced in the former (for example 4 or 8 times)?

In my opinion, a set of new results that clearly distinguish the likeliness of 'Wnt/Fzd7 feedback' and 'Wnt-independent Fzd7 expression' models are required. Or the conclusions based on the computational study should be modified according to the results.

The results of *in vivo* experiment show that Wnt/Fzd7 feedback regulates Fzd7 expression. Based on this, possibility of Wnt-independent Fzd7 expression should be discussed.

---

## [Author Response]

Essential revisions:1) Addition of data quantification, model explanation and discussion of the results are needed.

We greatly appreciate all the comments. According to your suggestions, we revised the manuscript as follows in detail.

2) The manuscript would greatly benefit from English language editing.

We are very sorry for the inconvenience. To make the texts more comprehensible, the revised manuscript was edited by a professional English editor, Dr. Steven D. Aird.

Reviewer #1 (Recommendations for the authors):Addition of data quantification, model explanation and discussion of the results would significantly improve the paper.

Thank you for the comment. We revised them as described above (graphs and statistical analyses in Figures1, 2, 4; Figure 1—figure supplement 1; Figure 3—figure supplement 7; Figure 4—figure supplement 1, 2).

Reviewer #2 (Recommendations for the authors):Mathematical analysisThis type of mathematics is easy enough, and experimental biologists can do the analysis by themselves. Alternatively, if the authors cannot handle this, replace all the "mathematical" in the manuscript with "computational".

Thank you for your comments. As we mentioned above, we performed the simulation with a numerical method to analyze a complicated model.

We revised “mathematical analysis” to “computational” or “numerical” as you suggested. (for instance, page 6, line 17).

CitationThe reference the authors cited (Umulis et al., 2009) for determining reaction parameters is a review article (page 30). The authors should cite the original measurement (Hatta 2000 10.1002/1097-0282(2000)55:5<399::AID-BIP1014>3.0.CO;2-9, Iemura 1998 10.1073/pnas.95.16.9337.)

Thanks. We now cited original articles for the parameters used in this revision (page 20, line 3).

Reviewer #3 (Recommendations for the authors):Abstract: Authors need to clearly distinguish between results from experiments and results from simulations.For example, the reader cannot determine whether the following results are derived from experimental data or from simulations.In addition, the result presented in the next sentence, "We found that…by restricting sFRP1 spreading" is quite ambiguous. The reader may mistakenly believe that all the data were obtained by experiments in the cardiac field.

We agree with all the points. We changed the descriptions.

Lines 8-9, "We revealed…that this receptor-feedback is essential for shaping a steep gradient of Wnt".

We added “with a combination of experiments and mathematical modeling”

Lines 9-10, "The feedback imparts robustness against fluctuations of Wnt ligand production and allows the system to reach a steady state quickly".

We added “computer simulation revealed that”

"We found that…by restricting sFRP1 spreading" is quite ambiguous. The reader may mistakenly believe that all the data were obtained by experiments in the cardiac field.

We performed a new experiment to examine the necessity of N-acetyl HS in the cardiac field as described above. So, this description was not changed in this revision.

Figure 1C, F It is quite difficult to determine whether the expression of fzd7 in the prospective heart file is affected or not. It would be better to add magnified images. Also, staining with gata5, a pericardial marker, can help identify this field in embryos.

Thank you. We added magnified images and added new results of staining with gata5 gene (Figure 1D).

Figure 2: Adding some negative control, for example, secreted mVenus (without Wnt), can increase the reliability of this result.

Thank you. We performed an experiment with secreted mVenus as a negative control and quantified the fluorescence intensities (Figure 2; page 6, line 7).

Fluorescence of secreted mVenus is barely observed on the cellular membrane because unbound ligands cannot be observed (Mii et al., 2021). In the present experiment, the secreted mVenus was confirmed to have been secreted because the fluorescence was clearly detected with anti-GFP antibody (morphotrap) expression (Figure 2—figure supplement 1C).

Supplemental Figure 6E: Analysis of the expression of marker genes in the pericardium and myocardium would improve the quality of this result.

Thank you. We analyzed the expression of other marker genes, gata5 (pericardium) and tnni3 (myocardium). Consistent with the fzd7 expression, gata5 expression was expanded and tnni3 expression was restricted.

We added these explanations (page 9, line 7; Figure 3—figure supplement 7D).

Figure 4C, D: In Figure 4C, it is unclear whether N-acetyl HS is expressed in the prospective myocardium or not. Furthermore, the distribution of HS shown in Figure 4D is puzzling to me. Although the text explains that N-acetyl HS is expressed predominantly in the outer part of the cardiogenic mesoderm, in this figure HS is present in both the pericardium and myocardial regions. Probably this is due to an inadvertent mistake by the author, but I think this figure should be corrected.

Thank you. As you mentioned, the expression of N-acetyl HS (previous Figure 4C) in the prospective myocardium is quite low: the signals are mainly on the prospective pericardium. Accordingly, we corrected the schematic figure (Figure 4H). We also revised the text (page 10, line 20) for more clarity.

[Editors' note: further revisions were suggested prior to acceptance, as described below.]

The manuscript was considerably improved compared with the previous version addressing the reviewer's comments. However, I feel that the conclusions authors are trying to make solely from the computational results are too strong.The authors compare two alternative models that ensures Fzd7 expression that causes thin Wnt activation domain: Wnt/Fzd7 feedback and Wnt-independent Fzd7 expression. The authors argue that the results of computer simulations prefer the former model. However, I have an impression that both models are possible. The authors have to show a set of new computational results that clearly demonstrate that the former model is likely. Or the text should be modified so that both models are preferred according to the computational study. The results of *in vivo* experiment expressing TCF-dnFzd7 indicates that Wnt/Fzd7 feedback regulates Fzd7 expression. Based on this, possibility of Wnt-independent Fzd7 expression may be discussed. It's common in biology that multiple mechanisms cooperate to ensure robustness.In Page 7, line 25-29, two alternative conditions 'no feedback and high expression' and 'with feedback and low expression' are compared. The magnitude of feedback and initial expression have to be indicated in the main text and figure.

We revised the text and figure (page 7, line 168; page 8, line 195; Figure 3A-B).

The Figure 3B legend says that the results are the same as Figure 3—figure supplement 2A. I don't think they are the same. Fix this point.

Considering your comments below, Figure 3B is deleted. However, we would like to answer your question here; the X-axis is different in Figure 3B and Figure 3—figure supplement 2A but the other points such as Y-axis and the values for the simulation were the same. We are sorry for the confusion caused by our poor explanation.

The authors argue that the latter is more robust. But, what happens if the initial expression is further enhanced in the former (for example 4 or 8 times)?

As you may have suggested, when the initial expression is higher, the system is robust. The high initial expression has the same ability as the feedback expression of *fzd7* (page 7, line 182).

In my opinion, a set of new results that clearly distinguish the likeliness of 'Wnt/Fzd7 feedback' and 'Wnt-independent Fzd7 expression' models are required. Or the conclusions based on the computational study should be modified according to the results.

As we mentioned above, we agree with you thinking that both models are possible. Thus, we modified the conclusion of the computational study (page 7, line 184). In addition, we have deleted previous Figure 3B, which emphasized the importance of feedback.

The results of *in vivo* experiment show that Wnt/Fzd7 feedback regulates Fzd7 expression. Based on this, possibility of Wnt-independent Fzd7 expression should be discussed.

Thank you for the critical comments.

According to the *in vivo* experiment, the high expression of *fzd7* was suggested to be mainly via Wnt/Fzd7 feedback loop, and Wnt-independent *fzd7* expression was suggested to be minor. We added details of this explanation to page 13, line 328.